# Concentration-dependent effect of plant secondary metabolites on bacterial and fungal microbiomes in caterpillar guts

Hana Šigutová,[1,2] Petr Pyszko,[1] Martin Šigut,[1,3] Kateřina Czajová,[1] Martin Kostovčík,[3] Miroslav Kolařík,[3] Denisa Hařovská,[1,3] Pavel Drozd[1]

**ABSTRACT** Plant–herbivore interactions have been modulated by plant secondary metabolites (PSM), which exert a strong pressure on herbivore microbiome. This study examined the effects of different PSM types and concentrations on caterpillar fitness, composition of gut bacterial and fungal assemblages, and microbiome network stability and symbiotic bonds in relation to the caterpillar diet breadth. Polyphagous and monophagous caterpillars sampled from oak were reared on an artificial diet (AD) containing PSM native (tannic acid) and non-native (tannivin and salicylic acid) to oak at varying concentrations, along with control treatments (starving and fed by oak leaves or AD without PSM). Their gut microbiome was profiled using 16S and ITS2 rRNA gene metabarcoding. Contrary to expectations, the diet breadth combined with the PSM type had no effect on weight gain. The bacterial composition was shaped by PSM concentration, while caterpillar species and diet breadth had no effect. Compared with bacteria, concentration had no effect on the fungal composition, which was more influenced by diet breadth than by caterpillar species. Leaf-fed caterpillars harbored the highest microbial richness. In AD-fed caterpillars, bacteria formed more complex networks than fungi, and the complexity was simplified with higher PSM concentrations. We identified taxa significantly associated with caterpillar guts. Notably, the association *Lactobacillus–Lactococcus–Streptococcus* was universally present across all caterpillar species, regardless of diet breadth. Our findings emphasize the importance of considering PSM concentration and composition in understanding caterpillar–gut microbiome interactions. Further research will validate the functional roles of identified microbial taxa and their significance for caterpillar hosts.

**IMPORTANCE** The caterpillar gut is an excellent model system for studying host–microbiome interactions, as it represents an extreme environment for microbial life that usually has low diversity and considerable variability in community composition. Our study design combines feeding caterpillars on a natural and artificial diet with controlled levels of plant secondary metabolites and uses metabarcoding and quantitative PCR to simultaneously profile bacterial and fungal assemblages, which has never been performed. Moreover, we focus on multiple caterpillar species and consider diet breadth. Contrary to many previous studies, our study suggested the functional importance of certain microbial taxa, especially bacteria, and confirmed the previously proposed lower importance of fungi for caterpillar holobiont. Our study revealed the lack of differences between monophagous and polyphagous species in the responses of microbial assemblages to plant secondary metabolites, suggesting the limited role of the microbiome in the plasticity of the herbivore diet.

**KEYWORDS** plant secondary metabolite, tannin, tannivin, salicylic acid, network stability, invertebrate–microbe interactions, bacterial and fungal microbiomes

Address correspondence to Petr Pyszko, petr.pyszko@osu.cz, or Pavel Drozd, pavel.drozd@osu.cz.

The authors declare no conflict of interest.

See the funding table on p. 15.

During coevolution between plants and phytophagous insects, plant–herbivore interactions have been mediated by plant secondary metabolites (PSM) (1). These substances profoundly reduce the plant consumer performance through antifeeding effects (2, 3). Besides detoxification enzymes encoded within the insect genome, gut microbiome members, mainly bacteria and fungi, are involved in many metabolic pathways helping their host digest toxic plant compounds, thus imparting resistance to PSM (4–7). The microbiome genome complements the herbivore genome, forming an evolutionarily adaptive holobiont (8, 9).

Lepidoptera represent one of the most diverse insect orders (10), and their caterpillars are deleterious agricultural and forest pests worldwide. Unlike other insect groups, such as beetles or aphids, which harbor diverse microbial communities (4, 11), the presence and functional importance of caterpillar gut microbiome members remain a subject of ongoing debates (12–15) due to the absence of specialized microbe-housing structures and rapid food passage through the gut (12). However, caterpillars can harbor species-specific microbial consortia, particularly bacteria (14, 16). Moreover, even environmentally acquired microbes can play a significant role in PSM degradation (17).

Tannins are the most widespread PSM (18). These complex polyphenols are divided into condensed and hydrolyzable molecules (19) and are toxic to a wide range of herbivorous insects (20, 21). Hydrolyzable tannins pose a greater health risk as they can oxidize in insect guts to reactive oxygen species (22, 23), thereby accelerating damage to DNA, proteins, or carbohydrates (24). In alkaline caterpillar midgut, the prooxidant effect is enhanced (25). Tannins can be degraded by various bacteria and fungi [(reviewed in reference (26)].

While the impact of tannins on caterpillar growth, survival, or gut physiology has been extensively studied (21, 25, 27, 28), the effect on their gut microbiome has been neglected [but see references (29, 30)]. Significant changes in gut microbiome composition have been reported under tannin exposure in larval insects (29, 31) and vertebrates (32), including humans (33). To deepen our understanding of the role of tannins in plant–herbivore interactions, manipulative experiments controlling tannin levels are needed, as most studies have relied on correlations (34). Differentiating between major tannin types is also crucial (25). Previous studies on the effect of controlled PSM levels on insect herbivore microbiomes have been limited to bacteria and single polyphagous species on an artificial diet (29, 35) or *in vitro* studies targeting specific gut microbiome members (36, 37), neglecting the fungal component and the effect of diet breadth.

Diet breadth refers to the variety or range of different host plants or food sources that a particular species can or does consume, reflecting the extent to which a herbivore is specialized or generalized in its feeding habits. Unlike specialized feeders, polyphagous species may face more diverse exposure to PSM, influencing the establishment of their microbiome and their ability to utilize the plant substrate (38). Consequently, polyphagous and monophagous species differ in their detoxification capacity (39). The gut microbiome plays a crucial role in host adaptation to the dietary niche in relation to the presence of specific PSM (5, 40, 41). The PSM-induced changes in the gut microbiome structure may be considered as initial step toward host plant specialization (37). Investigating changes in the gut microbiome of polyphagous and monophagous species under varying concentrations of PSM native and non-native to host plants may help clarify the role of the microbiome in herbivore diet plasticity and, ultimately, host plant specialization.

We studied the effect of PSM on caterpillar fitness and gut bacterial and fungal assemblages in relation to diet breadth. We sampled caterpillars of polyphagous and monophagous lepidopteran species from oak and reared them on an artificial diet enriched by microorganisms and containing the following substances: (i) hydrolyzable tannic acid native to oak (19), (ii) quebracho tannivin or non-native condensed tannin, or (iii) salicylic acid (native to willows). The PSM concentrations varied to account for the concentration-dependent effect (5, 42). We expected significant differences in microbiome composition based on tannin concentrations and the substance type due

to differences in their digestion mechanism (see 43). We hypothesized that microbiome of all species, regardless of diet breadth, would contain bacterial and fungal consortia able to degrade tannin as they were sampled from oaks. Contrarily, we assumed that caterpillars on non-native tannins and more markedly on a salicylic acid-containing diet would lack specific microbial groups. However, the generalist microbiome would have higher plasticity to adapt to the novel diet compound. We also analyzed the stochasticity of processes shaping microbiome assembly and stability of network connections that can be disrupted after stress exposure (44).

## RESULTS

### Microbial quantification

The gut bacterial load exceeded the fungal load (median = 267,648 cells/g, interquartile range [IQR] 23,363–1,119,731; median = 50.1 cells/g, IQR 12.5–265.8, respectively) and was affected by PSM concentration, depending on the compound type (Table S4; Fig. S1).

### Impact on fitness

Starved caterpillars had always negative weight gain. In AD-fed caterpillars, the weight change was species specific (df = 327, $F$ = 20.39, $P$ < 0.001; Fig. 1a), depended negatively on the concentration of PSM (df = 336, $F$ = 4.48, $P$ = 0.004; Fig. 1b), and differed between compound types (df = 334, $F$ = 9.18, $P$ < 0.001; Fig. S2), where larvae fed by salicylate had a lower weight gain ($t$ = −4.62, $P$ < 0.001). The effect of diet breadth (df = 340, $F$ = 0.69, $P$ = 0.290) and its interaction with the compound type were not significant, neither for salicylate ($t$ = 1.22, $P$ = 0.224) nor tannin ($t$ = −0.27, $P$ = 0.789) or tannivin ($t$ = −1.30, $P$ = 0.196).

### Richness

The number of reads for bacteria and fungi is described in Fig. S3. The rarefied/extrapolated bacterial richness was 21.05 ± 0.57 genera per 1,000 reads for caterpillars and 13.75 ± 0.77 for diet. Caterpillar bacterial richness primarily correlated with fungal richness (df = 339, $F$ = 39.74, $P$ < 0.001), was affected by the concentration level (df = 337, $F$ = 12.27, $P$ < 0.001; Fig. 2a), and varied among species (df = 327, $F$ = 11.40, $P$ < 0.001; Fig. 2b; Fig. S4), but the diet breadth had no effect ($t$ = 0.66, $P$ = 0.511). The effect of the compound type was not significant (df = 334, $F$ = 2.07, $P$ = 0.104), without interaction with diet breadth (df = 324, $F$ = 1.04, $P$ = 378). Bacterial richness differed significantly between AD-fed, leaf-fed, and starved individuals (df = 410, $F$ = 21.33, $P$ < 0.001).

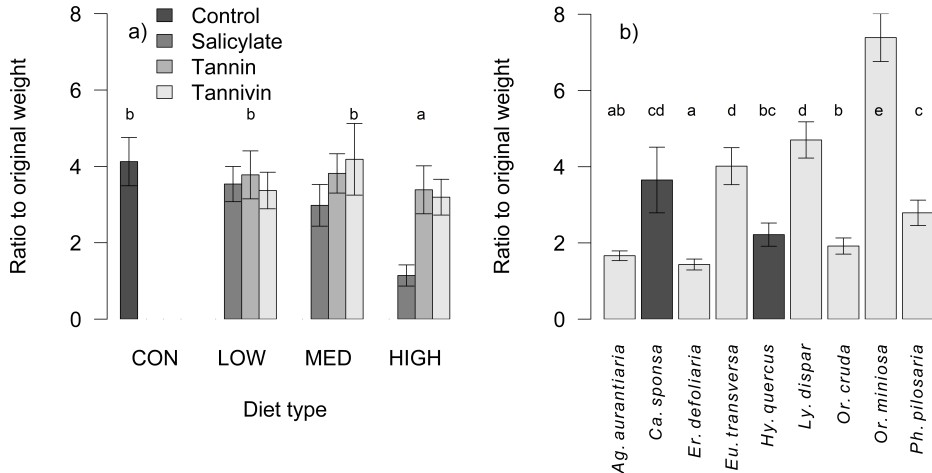

**FIG 1**  Ratio between initial and final weight for AD-fed caterpillars (a) in relation to compound types and concentration levels, (b) among species (mean ± SE). Individual letters (a, b, c, d) indicate groups that are significantly different from each other according to the main factor displayed on the x-axis. (a) Diet type and (b) caterpillar species.

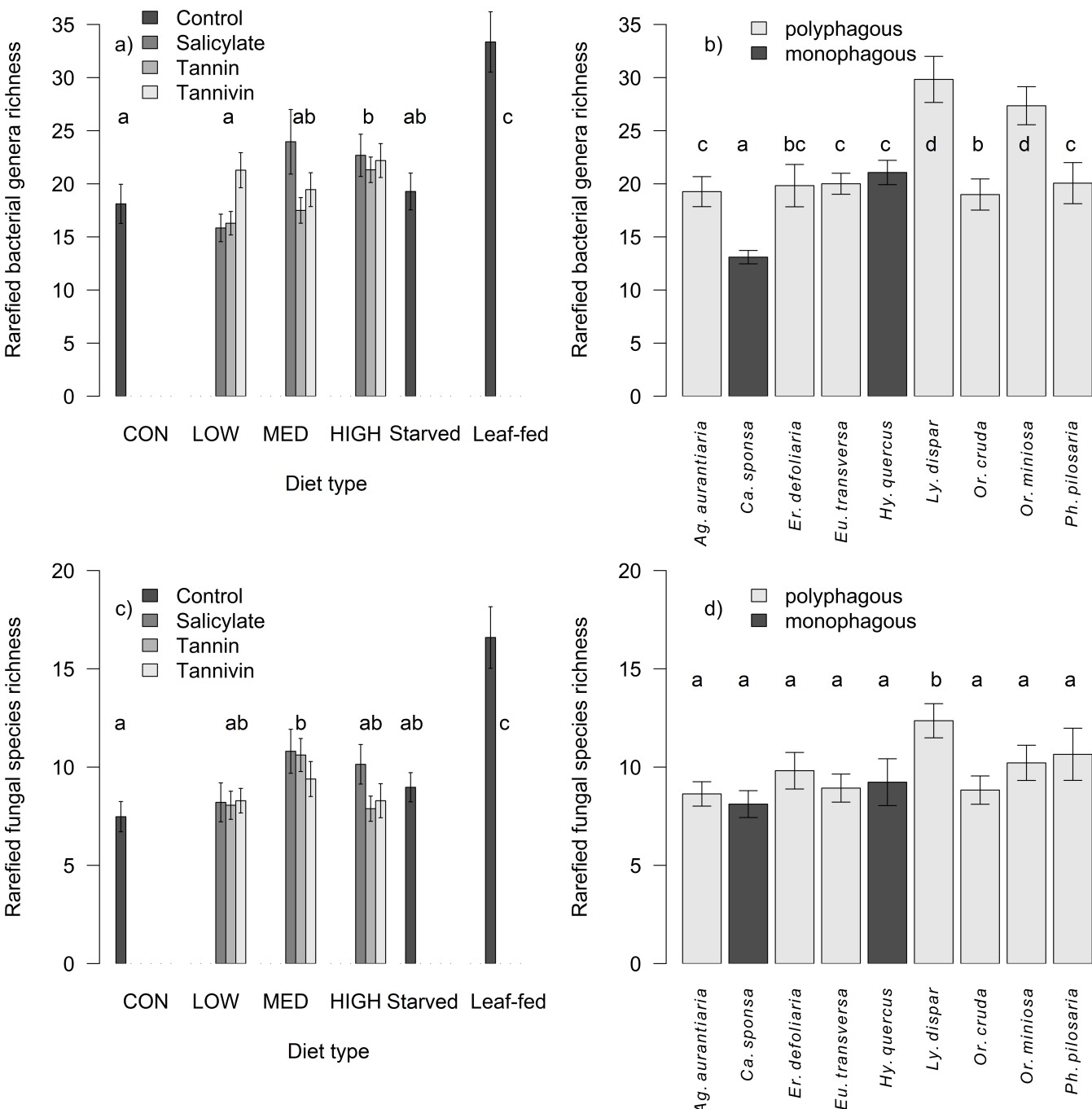

**FIG 2** Differences in rarefied bacterial genera richness (a, b) and fungal species richness (c, d) (mean ± SE) in relation to individual concentration levels and compound types (a, c) and among caterpillar species (b, d). Individual letters (a, b, c, d) indicate groups that are significantly different from each other according to the main factor displayed on the x-axis.

The rarefied/extrapolated fungal richness was 9.63 ± 0.30 species per 200 reads for caterpillars and 10.51 ± 0.73 for AD. Caterpillar fungal richness was primarily correlated with bacterial richness (df = 339, F = 23.42, P < 0.001) and showed an effect of concentration, peaking at the medium level (df = 337, F = 3.55, P = 0.030; Fig. 2c). The effect of compound type (df = 334, F = 1.63, P = 0.182), diet breadth (t = −0.22, P = 0.829), their interaction (df = 323, F = 1.69, P = 0.168), and host species (df = 326, F = 1.13, P = 0.343; Fig. 2d; Fig. S5) was not significant. Richness differed significantly among AD-fed, leaf-fed, and starved individuals (df = 410, F = 25.63, P < 0.001). The bacterial and fungal richness

showed a strong positive correlation ($\rho$ = 0.185, $S$ = 5385507, $P$ < 0.001). Factors that affected AD richness are explained in Fig. S6.

## Similarity

The bacterial similarity of AD-fed caterpillars to their diet depended positively on the concentration level (df = 339, $\chi^2$ = 6.23, $P$ = 0.013; Fig. 3a) but not on the caterpillar species (df = 328, $\chi^2$ = 5.46, $P$ = 0.604; Fig. 3b), diet breadth (df = 338, $\chi^2$ = 0.30, $P$ = 0.582), or compound type (df = 335, $\chi^2$ = 0.65, $P$ = 0.886). There was no interaction between diet breadth and compound type (df = 325, $\chi^2$ = 0.65, $P$ = 0.966). The fungal similarity was not influenced by concentration (df = 339, $\chi^2$ = 3.23, $P$ = 0.072; Fig. 3c), caterpillar species (df = 328, $\chi^2$ = 2.41, $P$ = 0.934; Fig. 3d), diet breadth (df = 335, $\chi^2$ = 0.19, $P$ = 0.667), compound type (df = 336, $\chi^2$ = 3.20, $P$ = 0.361), or its interaction with diet breadth (df = 325, $\chi^2$ = 0.57, $P$ = 0.904).

## Composition

A small but significant amount of the variation in bacterial composition in AD-fed individuals was explained by caterpillar species (8.45% of variability; df = 327, $F$ = 4.57, $P$ = 0.001; nested in diet breadth that further explained 1.36% of variability; df = 339, $F$ = 5.14, $P$ = 0.001), concentration level (3.10%, df = 336, $F$ = 3.91, $P$ = 0.001), and compound type (1.01%, df = 334, $F$ = 1.92, $P$ = 0.011), with no interaction between diet breadth and compound type (df = 324, $F$ = 0.73, $P$ = 0.875). Monophages had higher $\beta$-diversity than polyphages (df = 339, $F$ = 4.43, $P$ = 0.036). Taking concentration as a focal explanatory variable, the groups differed in $\beta$-diversity (df = 337, $F$ = 8.58, $P$ < 0.001). The individuals fed a high PSM concentration diet had a lower $\beta$-diversity than those fed by a control ($P$ = 0.010), low ($P$ < 0.001), or medium concentration diet ($P$ = 0.001). The P-CCA comparing the individuals with leaf-fed and starved individuals showed significant differences between the groups (df = 400, $F$ = 4.60, $P$ = 0.001; Fig. 4a).

The variation in fungal composition of AD-fed individuals was explained by diet breadth (13.66%, df = 339, $F$ = 4.78, $P$ = 0.001), and nested caterpillar species (3.46%, df = 327, $F$ = 1.73, $P$ = 0.001), but the effects of the compound type and concentration were not significant (1.09%, df = 336, $F$ = 1.27, $P$ = 0.083; 0.73%, df = 334, $F$ = 1.27, $P$ = 0.112, respectively) nor was the interaction between compound type and diet breadth (0.80%, df = 324, $F$ = 0.93, $P$ = 0.626). Monophages and polyphages did not differ in $\beta$-diversity (df = 339, $F$ = 0.28, $P$ = 0.597). Taking concentration as a focal explanatory variable, the groups did not differ in $\beta$-diversity (df = 337, $F$ = 1.48, $P$ = 0.219). P-CCA comparing AD-fed, leaf-fed, and starved individuals showed significant differences between groups (df = 400, $F$ = 3.10, $P$ = 0.001; Fig. 4b). Factors shaping the composition of AD are explained in Fig. S7.

## Community assembly and symbiotic bonds

Individuals fed by AD without PSM and starving had the highest stochasticity of bacterial community assembly, while those fed by low-PSM concentration level diet had the lowest stochasticity. The stochasticity tended to increase in individuals fed by AD with medium and high concentrations of PSM (df = 7, $\chi^2$ = 3.54, $P$ = 0.060). The effect of the compound type was not significant (df = 5, $\chi^2$ = 1.74, $P$ = 0.420, Table S5). In fungi, neither the effect of concentration nor the compound type was significant (df = 5, $\chi^2$ = 0.02, $P$ = 0.889; df = 5, $\chi^2$ = 3.43, $P$ = 0.794; respectively, Table S6). The bacterial assemblages were assembled less stochastically than the fungal ones (df = 1, $\chi^2$ = 7.77, $P$ = 0.005, Fig. S8).

Analysis of threshold indicator taxa identified 16 bacterial and 5 fungal taxa with significant changes in relative abundances with increasing levels of PSM. As PSM concentrations increased, relative abundances of 12 bacterial and 2 fungal taxa increased, while 4 bacterial and 3 fungal taxa decreased (Table 1; Fig. S9 and S10).

More bacteria than fungi were included in the networks. The most complex networks were observed for leaf-fed individuals. The network complexity was abrupted in AD-fed individuals, and with increasing concentrations of PSM, the networks were simplified.

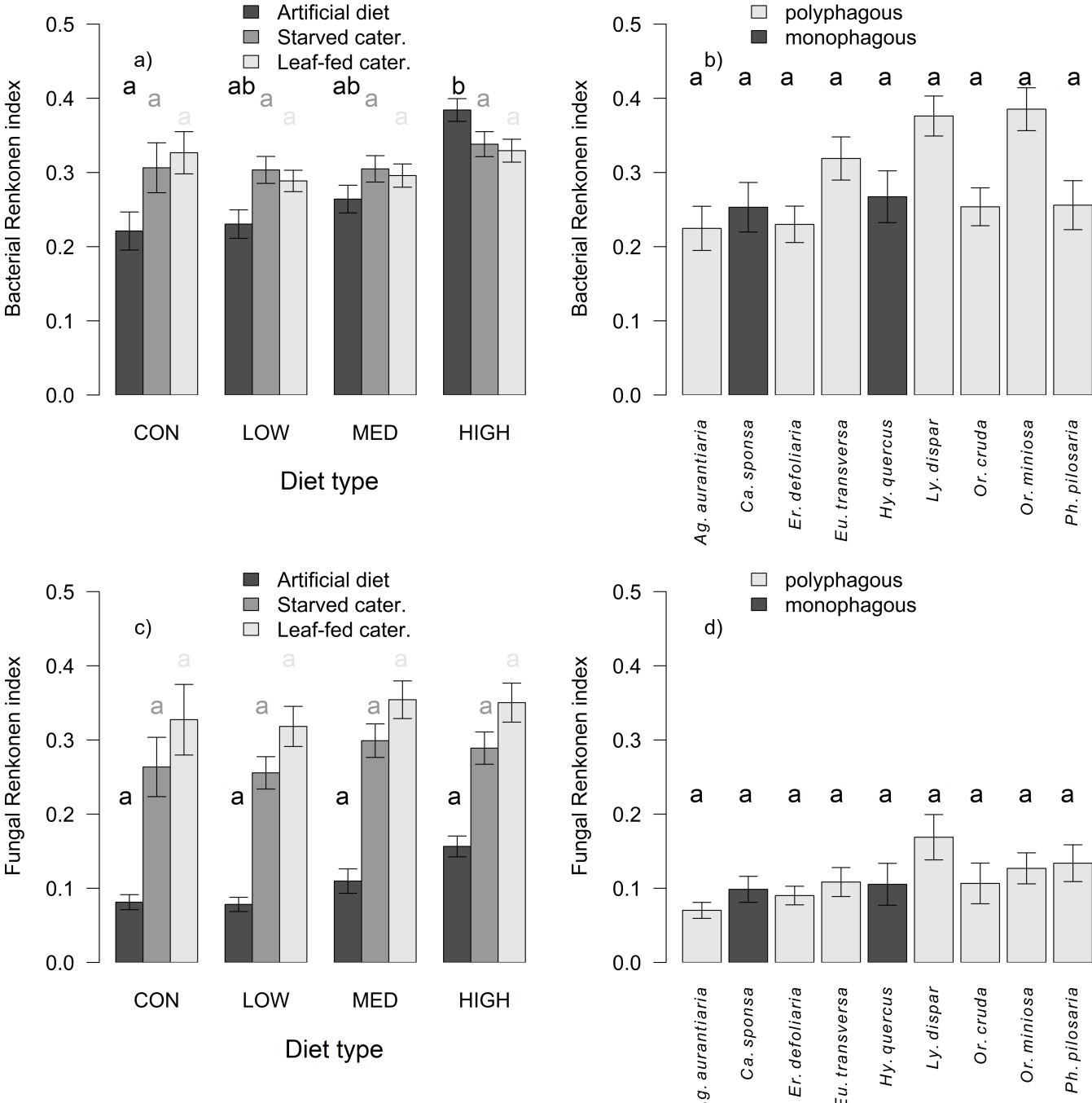

**FIG 3** Differences in Renkonen similarity index between AD-fed caterpillars and their AD (mean ± SE) for bacterial (a, b) and fungal microbiota (c, d) among caterpillar species (b, d) and in relation to individual concentration levels (con—without PSM, low, medium, high) (a, c). The subplots a and c show also the comparison of AD-fed individuals with starved and leaf-fed individuals. Individual letters indicate groups that are significantly different from each other according to the main factor displayed on the x-axis. For (a and c), individual groups on the x-axis (diet type) were compared separately according to the level of the factor indicated in the legend.

*Lactobacillus*, *Lactococcus*, and *Streptococcus* were among taxa that persisted at high PSM concentrations, supplemented at lower concentrations by *Cutibacterium* and *Undibacterium* (Fig. S11). The networks at the species level revealed a universal association of *Lactobacillus*, *Lactococcus*, and *Streptococcus*, sometimes accompanied by other taxa (Table S8).

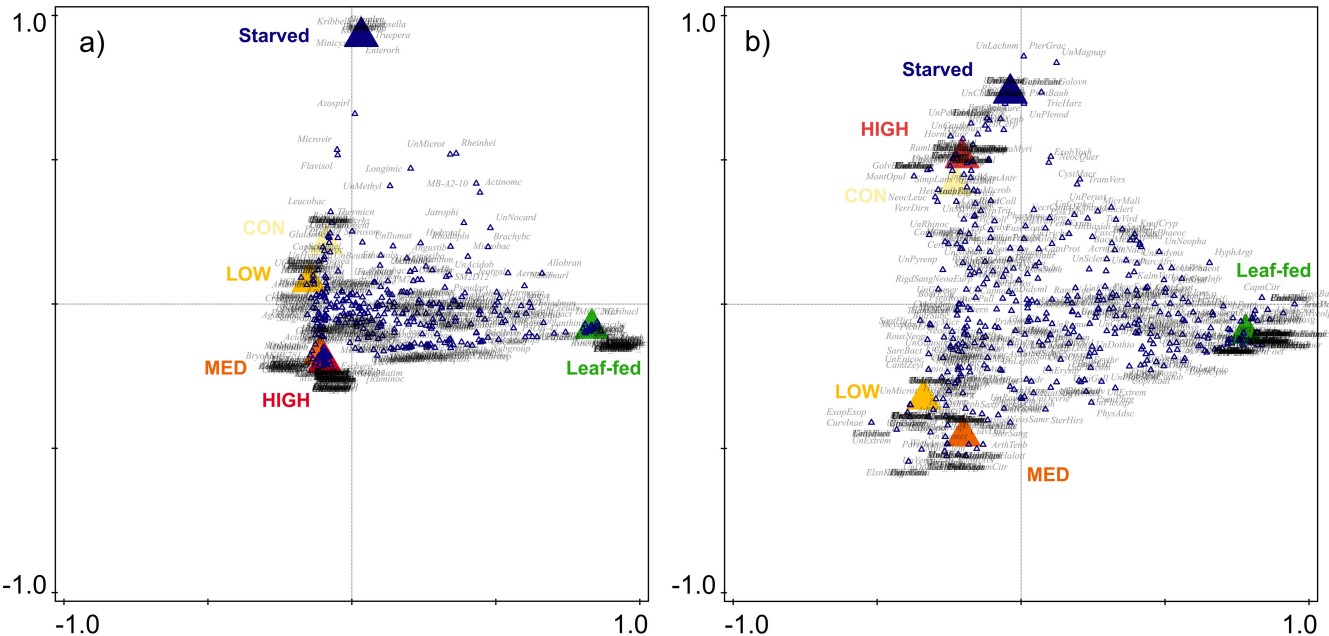

**FIG 4** Partial canonical correspondence analysis (p-CCA) plots showing dissimilarity in the composition of (a) bacterial (df = 400, F = 4.60, P = 0.001) and (b) fungal microbiota (df = 400, F = 3.10, P = 0) of leaf-fed and AD-fed caterpillars on control (CON), low-concentration (LOW), medium-concentration (MED), and high-concentration (HIGH) diet.

According to the iVikodak analysis using the KEGG database, the functional profiles of leaf-fed and AD-fed individuals on the control AD differed from those of starved and AD-fed individuals with PSM. The former showed greater activity in the bacterial metabolic pathways involved in cellular processes, environmental information and genetic information processing, metabolism, organismal systems, and human diseases (P ≤ 0.01; Fig. 5). Regarding only individuals fed AD with PSM, they had lower metabolism rates and did not differ from each other.

## DISCUSSION

The weight gain in AD-fed caterpillars was species specific and varied between substances. No effect of diet breadth was an unexpected finding, as specialists have better physiological adaptations to overcome plant defenses compared with generalists (45). However, the tolerance of specialists is typically limited to low toxin concentrations and diminishes at higher levels (46). This was corroborated by our study; weight gain decreased with increasing PSM concentration. Nevertheless, its interaction with specialization was not significant. Individuals fed by salicylate-enriched AD exhibited the lowest weight gain. Host intrinsic detoxification capacity may be supplemented by host plant-derived microbes (41, 47). The experimental individuals were sampled from oaks; therefore, they may have lacked specific microbial groups that helped them deal with salicylates that are dominant PSM in Salicaceae (48).

The environmental acquisition of key microbial groups for the detoxification of PSM can be confirmed by the fact that the compound type played an important role in shaping the bacterial composition of AD-fed caterpillars. Moreover, bacteria and fungi in AD-fed individuals differed from starved and leaf-fed individuals, highlighting the effect of PSM on microbial consortia. Nevertheless, the effect of the compound type was secondary compared with the effect of host species as the primary driver of bacterial composition and richness, while the effect of diet breadth was insignificant. Although numerous studies have emphasized the importance of diet over caterpillar identity in shaping bacterial assemblages (47, 49), recent studies have shown that host species is one of the most important factors (14, 16). Concentration also played a significant

**TABLE 1** Threshold indicator taxa analysis for AD-fed larvae with an increasing concentration level of PSM[a]

| Community | Taxon | freq | IndVal | bsiv.prob | zscore | purity | reliability | trend |
|---|---|---|---|---|---|---|---|---|
| Bacteria | *Corynebacterium* | 138 | 33.12 | 0.040 | 6.80 | 1.00 | 0.99 | Increasing |
| Bacteria | *Lawsonella* | 31 | 12.11 | 0.040 | 4.43 | 0.96 | 0.99 | Increasing |
| Bacteria | *Kocuria* | 123 | 28.68 | 0.040 | 6.22 | 1.00 | 1.00 | Increasing |
| Bacteria | *Micrococcus* | 176 | 38.14 | 0.040 | 6.12 | 1.00 | 0.99 | Increasing |
| Bacteria | *Cutibacterium* | 330 | 62.89 | 0.040 | 7.86 | 1.00 | 1.00 | Increasing |
| Bacteria | unkn. Actinobacteria | 25 | 10.87 | 0.040 | 7.08 | 0.99 | 0.97 | Increasing |
| Bacteria | *Prevotella* | 38 | 14.82 | 0.040 | 7.76 | 0.98 | 0.98 | Increasing |
| Bacteria | unkn. Hymenobacteraceae | 5 | 16.51 | 0.040 | 15.02 | 1.00 | 0.96 | Decreasing |
| Bacteria | *Thermus* | 23 | 11.18 | 0.040 | 5.03 | 0.99 | 0.95 | Increasing |
| Bacteria | unkn. Bacillaceae | 300 | 62.00 | 0.040 | 9.06 | 1.00 | 1.00 | Increasing |
| Bacteria | *Lactobacillus* | 300 | 64.39 | 0.040 | 7.94 | 0.98 | 0.99 | Increasing |
| Bacteria | unkn. Clostridiaceae | 16 | 19.09 | 0.040 | 9.17 | 1.00 | 1.00 | Decreasing |
| Bacteria | *Undibacterium* | 203 | 41.04 | 0.040 | 6.22 | 0.99 | 0.97 | Increasing |
| Bacteria | *Erwinia* | 333 | 64.31 | 0.040 | 5.27 | 0.96 | 1.00 | Decreasing |
| Bacteria | *Acinetobacter* | 149 | 39.31 | 0.040 | 9.28 | 1.00 | 1.00 | Increasing |
| Bacteria | unkn. Proteobacteria | 7 | 11.09 | 0.040 | 10.04 | 1.00 | 0.98 | Decreasing |
| Fungi | *Alternaria* sp. | 162 | 32.02 | 0.040 | 3.85 | 0.99 | 0.97 | Increasing |
| Fungi | *Aspergillus ruber* | 68 | 21.62 | 0.040 | 8.59 | 1.00 | 1.00 | Decreasing |
| Fungi | *Saccharomyces cerevisiae* | 33 | 12.52 | 0.040 | 5.21 | 0.96 | 1.00 | Decreasing |
| Fungi | *Candida zeylanoides* | 9 | 5.05 | 0.040 | 5.92 | 1.00 | 0.95 | Decreasing |
| Fungi | *Fusarium graminearum* | 26 | 9.98 | 0.040 | 6.00 | 1.00 | 1.00 | Increasing |

[a]Community analysis for bacterial or fungal taxa; Taxon, Taxon with the identified trend; freq, number of non-zero abundance values per taxon; IndVal, Dufrene and Legendre 1997 IndVal statistic; obsiv.prob, the probability of an equal or larger IndVal from random permutation; zscore, IndVal zscore; purity, proportion of replicates matching observed assignment; reliability, proportion of replicate obsiv.prob values ≤ 0.05; trend, the simplified relationship between relative taxon abundance and increasing concentration level. Taxa with decreasing trends are underlined.

role in shaping bacterial composition, consistent with findings in larval camelia weevils (37), bark beetles (5), or adult longhorn beetles (50). The importance of concentration was further supported by the observation that the bacterial microbiomes became more similar to their diet as the concentration level increased, while host species, specialization, or compound type had no effect on similarity.

Unlike bacteria, diet breadth had a stronger influence on the fungal community composition than host species, whereas the compound type and concentration level had no effect. Moreover, the fungal similarity was not affected by any of the variables. The limited effect of concentration or compound type was unexpected, considering the ability of fungi to degrade tannins [reviewed in reference (26)] or salicylates (51, 52). However, fungi are more influenced by environmental acquisition and less dependent on the identity of host species compared with bacteria (14, 16), which agrees with the lack of importance of caterpillar species in shaping the fungal community in our study.

Leaf-fed individuals exhibited the highest bacterial and fungal richness. A species-rich microbiome is generally more resilient to toxic substances and supports system functions and host health (53, 54). Exposure to PSM may disrupt microbial balance [i.e., cause dysbiosis (55)], similar to the effects of antibiotic treatment on some key microbial group (56, 57). Therefore, the transition to AD with altered PSM concentrations consistently decreased richness. Fungal richness was also affected by the concentration level but not by caterpillar species. The peak in medium-level concentration suggested that both high and low PSM concentrations disrupt fungal balance (55). Although higher microbial diversity is associated with the consumption of different types of food in insects and polyphages host higher diversity than monophages (58, 59), our study did not find any specialization-related differences in bacterial and fungal richness.

Interestingly, the bacterial and fungal richness of AD-fed caterpillars was positively correlated. This may suggest an interaction between both these components, which is well documented in vertebrates (60, 61). Gut bacteria are perceived as controlling agents

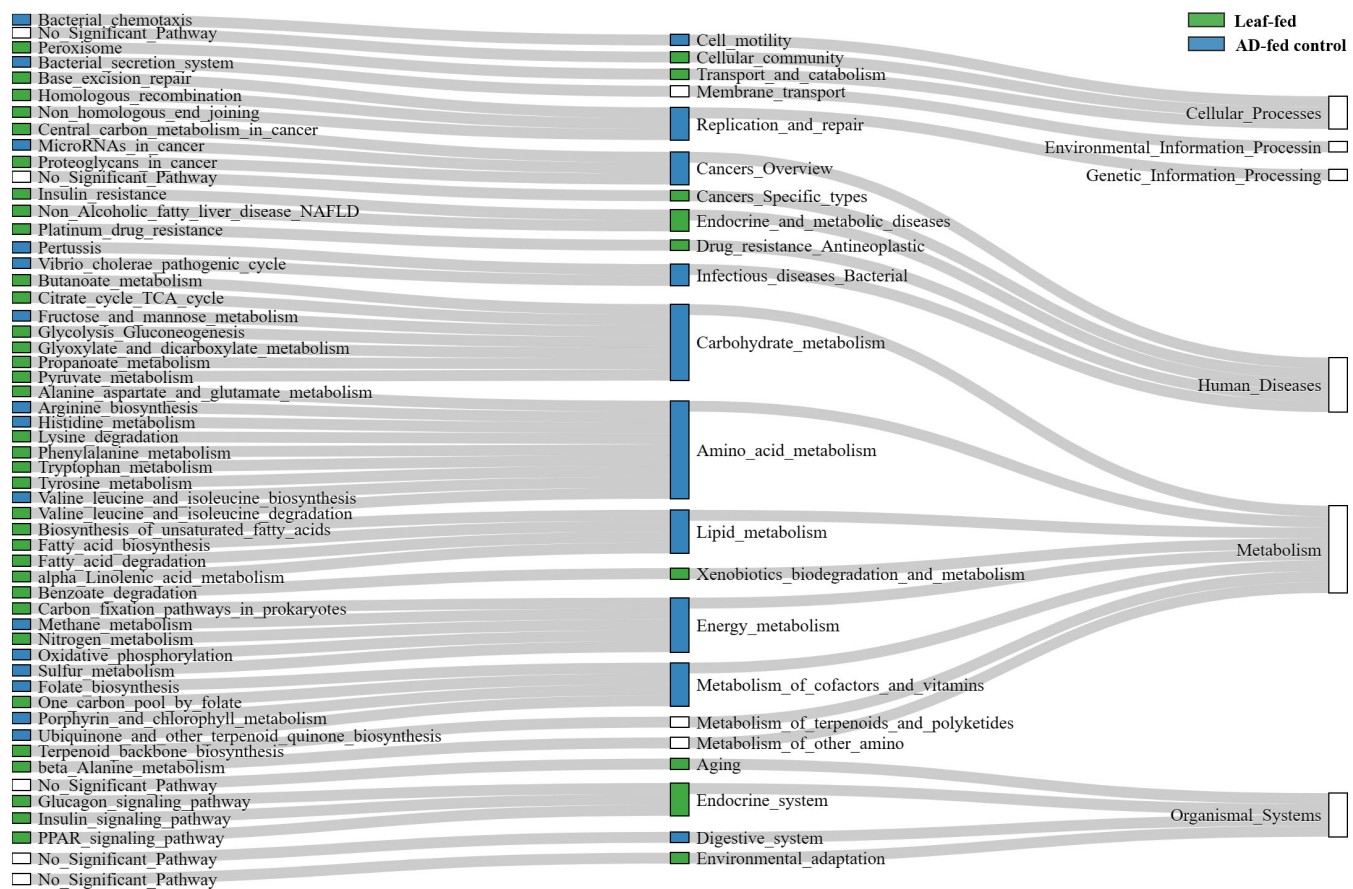

**FIG 5** Cladogram for the reconstruction of the most important significant pathways in leaf-fed and control AD-fed individuals ($P \leq 0.01$). Metabolic pathways of starved larvae and larvae fed by AD with PSM were not significantly increased in comparison with groups depicted.

due to their secretion of substances that regulate fungal growth (62), modulation of the host's immune system, or competition for adhesion sites and nutrients [reviewed in reference (61)]. Although elimination of a key bacterial group typically leads to fungal proliferation (61, 62), this may not be the case for caterpillars due to unique conditions of their midgut [e.g., extreme alkalinity; see reference (12)]. The median bacterial load in the guts of AD-fed caterpillars was approximately 5,000 times higher than that of the fungal one, suggesting the limited importance of fungi. Importantly, differences in biomass resulting from volume and size differences between bacteria and fungi must be taken into account (61); moreover, in caterpillars, functionally important taxa can occur in disproportionately low abundances (7). When considering the PSM concentration, the median bacterial load was approximately 12,500× higher than the fungal load at low levels and 41,000× higher at high levels, suggesting that bacteria outperformed fungi at high concentration levels.

Feeding on AD with a high concentration of PSM led to a significant decrease in bacterial β-diversity. Higher concentrations of PSM promote taxa with stronger metabolic capacity for toxins (5), leading to possible dominance of PSM-tolerant taxa and greater similarity between caterpillars fed a high PSM-concentration AD. Moreover, monophages had higher bacterial β-diversity than polyphages. Microbial disbalance may affect insect fitness and/or result in changes in β-diversity [reviewed in reference (44)], whereby greater dispersion may be an indicator of the dysbiotic state (63). Therefore, in specialists, the bacterial microbiome may be more disrupted by PSM than in generalists. Alternatively, this may result from the fact that both specialists, *Catocala sponsa* and *Hypaurotis quercus*, are phylogenetically more distant than the group of generalists.

Nevertheless, in fungi, the lack of specialization-related differences in β-diversity and the absence of the effect of concentration suggest that fungi are less important to their hosts than bacteria.

This was also corroborated by network analysis. In AD-fed caterpillars, bacteria were involved in more complex networks than fungi. The highest complexity was found in leaf-fed individuals. Switching to AD abrupted the complexity, and increasing PSM concentrations led to simplification of the network. However, the *Lactobacillus–Lactococcus–Streptococcus* association was always involved and persisted under high concentrations of PSM. These taxa are common inhabitants of insect guts (64–66) and can degrade tannins by producing tannase (67–69). Moreover, *Lactobacillus* promotes host immunity (70), while the genus *Staphylococcus* comprises taxa with protease activity (66) but also insect pathogens (70). Our results suggest that the *Lactobacillus–Lactococcus–Streptococcus* association may be especially meaningful for caterpillars, because such a specific interaction among commensal bacteria may inhibit pathogen infection, especially caused by fungi (71). At lower PSM concentrations, *Cutibacterium* and *Undibacterium* were involved. Although *Undibacterium* is associated mainly with aquatic environments, several species affiliated with this genus are involved in lipid metabolism (72, 73). *Cutibacterium* may help digestion (74).

In AD-fed caterpillars, we identified significantly associated taxa whose frequency changed with increasing PSM concentration. Among bacteria with increasing frequency were taxa commonly associated with insect guts, potentially beneficial for the host. Apart from *Cutibacterium* and *Undibacterium* described above, *Corynebacterium* produces amino acids (75) or tannase (76) and *Prevotella* and *Thermus* help to digest carbohydrates (77, 78), while *Acinetobacter* metabolizes toxic phenolic glycosides (6). *Lawsonella* associated with various insect taxa (79, 80) and *Kocuria* found in lepidopteran guts are both known as commensals and pathogens, but with limited information on their functional roles (81, 82). However, certain strains of *Kocuria* can degrade phenolic compounds (83). *Micrococcus* known as an insect pathogen (84) can also degrade cellulose (85). Both fungi with an increasing trend are plant pathogens; *Fusarium graminearum* is considered transient without any functional role in insects (86), while *Alternaria* dominates the guts under stress exposure (87) and inhibits caterpillar digestive enzymes (88). Taxa with a decreasing trend included common plant endophytes [*Erwinia* (89)] or insect pathogens [*Aspergillus* sp. (90); *Candida zeylanoides* (91)]. Nevertheless, *Aspergillus* may produce tannase (92). *Saccharomyces cerevisiae*, which may protect its host against fungal pathogens (93) and is involved in cellulose digestion, in association with cellulolytic bacteria (94), also showed a decreasing trend.

The importance of PSM concentration in shaping bacterial consortia was verified by neutral models. In leaf-fed individuals, the stochasticity was relatively low (78.85% taxa in the model prediction), but after the transition to AD or starvation, the stochasticity increased (83.78% and 84.76%, respectively), consistently with abruption of network connections. Surprisingly, low concentration levels of PSM decreased stochasticity, but increased concentration led to higher stochasticity. The effect of the compound type was insignificant, emphasizing the importance of concentration in shaping bacterial assemblages. In contrast, the fungal stochasticity for leaf-fed individuals was relatively high (89.90%), aligning with their high rate of environmental acquisition. Feeding on an AD free of PSM resulted in a decrease in fungal stochasticity (87.16%), and the concentration or compound type had no effect. Overall, bacteria exhibited less stochasticity than fungi, consistent with findings in leaf-mining caterpillars (16).

Analysis of KEGG pathways did not reveal any consistent pattern in bacterial metabolism related to PSM concentration, which was surprising, given its major effect on composition, similarity, richness, stochasticity, and bacterial networks. Leaf-fed and control AD-fed individuals showed higher activity in metabolic pathways involved in cellular processes, environmental and genetic information processing, metabolism, and organismal systems. Individuals on control AD displayed higher rates of pathways related to cell motility, replication, and repair but also metabolism of carbohydrates, amino acids,

lipids, energy, cofactors, and vitamins and processes related to the digestive system. This is consistent with the finding that the digestion and metabolism may be more efficient in caterpillars fed an artificial diet (58). Apparently, the efficiency decreases in the presence of PSM. Leaf-fed individuals had higher rates of pathways related to the cellular community, transport and catabolism, degradation of xenobiotics, processes of the endocrine system, and environmental adaptation. This result, along with network analysis and higher bacterial richness in leaf-fed individuals, suggests that a natural diet promotes healthy microbiome function.

Our study revealed the lack of differences in the PSM-induced responses of microbial assemblages between monophagous and polyphagous species, suggesting the limited role of the microbiome in the plasticity of the herbivore diet. However, contrary to previous studies proposing the lack of importance of the caterpillar gut microbiome (12, 95), we identified taxa possibly involved in metabolic processes in the caterpillar guts. Further functional verification using transcriptomics is needed to confirm their significance for the host. Furthermore, conducting pairwise *in vitro* experiments may help understand how complicated microbial assemblages interact inside the caterpillar gut under exposure to different levels of PSM. Consistently with our previous studies, we suggested the limited functional importance of fungi for caterpillar host.

## MATERIALS AND METHODS

### Experimental design overview

Our bioassays included three main PSM treatments: (i) tannic acid (TA), (ii) tannivin (TV, proanthocyanidin-rich tannin from quebracho, *Schinopsis lorentzii*), and (iii) salicylic acid (SA), each in three concentrations, and three control treatments: (i) starved caterpillars, (ii) oak leaf-fed caterpillars, and (iii) caterpillars fed by a control artificial diet (AD; without PSM). The caterpillars were sampled in the field (nine species) and reared on a randomly administered type of diet (treatment type). Individual caterpillars served as a replication unit (Table S1).

### Caterpillar sampling

The sampling was conducted in May 2018 in the temperate floodplain forest in central Moravia, Czechia (PLA Litovelské Pomoraví; 49.6932°N, 17.1399°E). Caterpillars of seven polyphagous and two monophagous species (Table S1) were sampled from *Quercus robur* manually or using 1 m$^2$ beating sheets. To minimize the impact of the developmental stage on the microbiome composition and diversity, we sampled only the 3rd–4th instar caterpillars. Each individual was captured using sterilized tweezers, transferred to a 1.5-mL centrifuge tube, and transported to the laboratory for a rearing experiment.

### Caterpillar rearing

Individuals from each species were weighed and reared on McMorran diet (96). To investigate the effect of PSM on caterpillar microbiome, the basic diet was enriched with (i) TA (Sigma-Aldrich, MO, USA), (ii) TV (Erblsöh, Geisenheim, Germany), and (iii) SA (Sigma-Aldrich, MO, USA), each at three concentrations: low, medium, and high (corresponding to 0.1%, 1%, and 10% for TA and TV and 0.1%, 0.33%, and 1% for SA, respectively). The medium concentration corresponded to the natural occurrence of these substances in tree foliage (34, 97). The concentration gradient was established as 10-fold in TA and TV but only 3-fold in salicylate because after adding SA in the concentration above 1%, the diet did not solidify and remained liquid. The concentration gradient was therefore adjusted in a way that the diet had always similar consistency, regardless of the substance added, to ensure it will be accepted by caterpillars. Caterpillars were reared for 10 days in plastic containers (one individual per container) containing an AD piece (approx. 400 mg) in a 16:8 h (light:dark) photoperiod at 25°C. During this period, the diet was restored and the containers were cleaned three times.

AD was prepared aseptically but the containers lacked filters, allowing for spontaneous microorganism enrichment, presumably from a homogeneous source. Starved individuals [group (i) in Experimental design overview] were kept in the containers without AD. Oak leaf-fed individuals [group (ii) in Experimental design overview] were killed instantly after being transferred to the laboratory, placed in a 1.5-mL centrifuge tube with 98% ethanol, and stored at −32°C for subsequent DNA isolation. Individuals from group (iii) in Experimental design overview were reared on the basic diet without PSM. At the end of the rearing experiment, each caterpillar was weighed, placed in a 1.5-mL centrifuge tube with 98% ethanol, and stored at −32°C for subsequent DNA isolation.

## Sample processing and DNA metabarcoding of bacteria and fungi

Caterpillars were surface sterilized, and the gut samples were prepared as described in Šigut et al. (14). DNA was extracted using a NucleoSpin Tissue DNA Isolation Kit (Macherey-Nagel, Düren, Germany) from approximately 200 mg of gut tissue following the manufacturer's protocol. Before cell lysis, the samples were repeatedly crushed in 1.5-mL tubes using pestles and liquid nitrogen. To ensure broad microbial diversity recovery and to avoid chloroplast recovery, we used highly degenerate primers. The bacterial V5–V6 16S rRNA region was amplified using 799F and 1115R (98, 99), and the fungal ITS2 rRNA gene region was amplified using ITS3_KYO2 and ITS4_KYO3 (100), with barcodes added to the 5′ end to enable sample identification. The whole process of amplification is described in Šigut et al. (14). All PCR products were checked using 1.5% agarose gel. To identify the bacterial and fungal composition of AD, we also processed 30 diet samples (100 mg per sample; 3 replicates of each concentration and treatment + 3 replicates of the diet without PSM) following the abovementioned protocol.

Triplicate PCR reactions of individual samples were pooled within each "plate library" (96 samples). Amplicons of specific lengths from individual libraries were extracted from the 2% agarose gel and purified using a QIAquick Gel Extraction Kit (Qiagen, Hilden, Germany). The DNA concentration was measured with a Qubit dsDNA BR Assay Kit (Thermo Fisher Scientific) and equalized to 20 ng/μL. The ligation of sequencing adapters and library-unique multiplex identifiers was performed using a KAPA Hyper Prep Kit. Libraries were quantified with the KAPA Library Quantification Kit (both Kapa Biosystems). We created one final library of bacterial samples and one of the fungal samples at 7.5 ng/μL by pooling the equimolar proportions of individual "plate libraries." Sequencing was performed at CEITEC (Masaryk University, Brno, Czech Republic) on NextSeq 500 for the bacterial library (1 × 150 bp reads) and on MiSeq for the fungal library (2 × 300 bp reads) (both Illumina Inc., San Diego, USA).

## Microbial quantification

To quantify microbial loads, we performed a qPCR analysis of the subset of DNA isolates ($n = 49$) representing a diverse range of caterpillar species, treatments, and concentrations. We used a Femto bacterial DNA quantification kit and a Femto fungal DNA quantification kit (both Zymo Research, CA, USA) following the manufacturer's instructions. The qPCR reactions were performed in duplicate using a CFX96 Real-Time PCR Detection System (Bio-Rad, CA, USA). Bacterial and fungal DNA concentrations were calculated against the standard curve through linear regression analysis and, subsequently, converted to the number of cells per 1 g of gut tissue.

## DNA metabarcoding data processing

Sequencing data were processed using QIIME 2.0 2020.2 (101). Raw reads demultiplexing and quality filtering were performed using the q2‐demux plugin. In fungal data sets, the ITS region was extracted using the q2-ITSxpress plugin (102). Subsequently, we used the DADA2 algorithm to denoise reads (103) and produced a feature table with counts of amplicon sequence variants (ASVs) per sample. To assign taxonomy, we used a trained naïve Bayes classifier against the SILVA_138_SSURef_Nr99 bacterial reference database

(104) and UNITE QIIME release for Fungi version 8.0 (105, 106) along with the q2‐feature‐classifier classify-sklearn (107). The resulting table contained 16,361,483 bacterial and 1,655,278 fungal reads represented by 8,615 and 3,055 ASVs, respectively. We identified contaminant ASVs using the "decontam" package (108) with the prevalence method using extraction controls as negatives (three per 96-well plate) and a probability threshold of 0.1 for the rejection of non-contamination. We discarded 164 bacterial and 35 fungal ASVs (0.62% of reads; Table S2) and removed reads associated with chloroplasts and mitochondria (2.41%) and those unassigned (3.23%). Finally, 15,253,031 bacterial and 1,635,659 fungal reads were used for analysis. An overview of the bacterial and fungal taxa, the number of reads, and the variables entering the analyzes is included in Table S3.

## Statistical analyses

### Impact on fitness

Data were analyzed in R 4.2.1 (109). The caterpillar weight change was calculated as the ratio between the initial and final weights and was compared among treatment groups using the generalized linear model (GLM) with Gamma distribution.

### Richness

ASVs were classified into genera for bacteria and species for fungi. Read counts were rarefied/extrapolated to the same depth (1,000 reads for bacteria, 200 reads for fungi) using the "iNEXT" library (110). Richness was analyzed using GLMs with Gamma distribution for caterpillars and using a linear model for AD. The final model was built by stepwise selection based on the Akaike information criterion (AIC). Explanatory variables included information about caterpillars (family, species, and diet breadth, with species nested within diet breadth), diet (compound type, concentration), the richness of the other microbiota component (fungi for bacterial richness and vice versa), and their interactions. Richness was compared among AD-fed, leaf-fed, and starved individuals by GLMs. Bacterial and fungal richness was compared using Spearman's rank correlation based on the model results.

### Similarity

We calculated the Renkonen quantitative similarity index based on relative abundances of reads (111) between the microbiome of AD-fed individuals and the following: (i) AD with the same concentration level and compound type, (ii) starved individuals of the same species, and (iii) leaf-fed individuals of the same species. We used GLMs with a binomial distribution (logistic link) to analyze the factors influencing differences in similarity between AD-fed caterpillars and their AD, including caterpillar species nested within specialization, compound type, concentration level, or their interactions. Furthermore, we compared the bacterial and fungal similarity of AD-fed individuals to AD, starved, and leaf-fed individuals using GLMs with the binomial distribution.

### Composition

We analyzed differences in the microbiota composition of AD-fed individuals by PERMANOVA with Bray–Curtis distance matrices (999 permutations), using the "vegan" library (112). The same explanatory variables as in richness analysis were used. We built the final model based on AICc through forward selection. To validate PERMANOVA results, we added the PERMDISP2 procedure to assess multivariate homogeneity of group dispersions (variances) using the Bray–Curtis distance and measuring distances to the group centroids (113). Differences in β-diversity between polyphages and monophages and between concentration levels were tested by ANOVA and Tukey HSD post hoc test. We also conducted partial canonical correspondence analysis (p-CCA) with all individuals, using individual groups (including leaf-fed and starved caterpillars) as

an explanatory variable and caterpillar species as a covariate (999 permutations). For the analysis of the composition of the microbiome of AD, we omitted the identity of caterpillar species as variable and replaced p-CCA with redundancy analysis (RDA). P-CCA and RDA analyses were performed in Canoco 5.01 (114).

### Community assembly and symbiotic bonds

To quantify the involvement of neutral processes in microbiome assembly, we fitted neutral models according to Sloan et al. (115) using the "reltools," "phyloseq," and "GUniFrac" libraries (116, 117, 118). Separate models were fitted for leaf-fed, starved, and AD-fed caterpillars for each concentration level and AD and for each species separately. To account for group size sensitivity, we randomly selected 30 samples per group (36 for the species-focused alternative). We rarefied samples to the same sequence depth (1,000 reads) before fitting the neutral models. We analyzed the ratio of taxa that fit the null models in relation to the concentration level and compound type (or diet breadth) using GLM with binomial distribution. We also compared the proportion for bacteria and fungi in leaf-fed individuals by Pearson's chi-squared test.

As the concentration level was more important than the compound type, we used the "TITAN2" library (119) to conduct a threshold indicator taxa analysis, employing the indicator value (IndVal) approach (120). This analysis identified taxa with significant changes in relative abundances as the PSM concentration levels increased. The partitioning was performed with a minimum split size of 4. Twenty-five replicates were used, and 100 replicates were employed during bootstrap resampling. The cutoff values for purity and reliability were set to 0.95.

We used the SparCC algorithm (121) to compute sparse correlations for compositional data. A cutoff value of ≥0.60 was used. The outer and inner loops consisted of 20 and 10 iterations, respectively. Kleinberg's hub centrality scores were calculated for the network, and superfluous vertices were removed. Densely connected subplots were identified using the walk trap algorithm with a random walk length of 4. The degree of a vertex (the number of adjacent edges) was analyzed using a power-law distribution function, and the fit of the function was assessed by the Kolmogorov–Smirnov test.

Furthermore, we inferred the functional characteristics of the bacterial part of the microbiome by the iVikodak platform (122). We utilized a Global Mapper module to obtain the functional profiles, with percentage normalization of data and median central tendency, using the ICo algorithm assuming independent contributions of bacteria. Microbiome functions were inferred based on KEGG categories (https://www.genome.jp/kegg).

### ACKNOWLEDGMENTS

We thank the members of the Laboratory of Insect Trophic Strategies for participation in field sampling, caterpillar rearing, and laboratory processing. This study was supported by the Czech Science Foundation (GA22-29971S).

H.Š. did the writing—original draft; P.P. did the conceptualization, methodology, investigation, formal analysis, visualization, and writing—review and editing; M.Š. did the investigation, methodology, data curation, and writing—review and editing; K.C. did formal analysis, investigation, and writing—review and editing; M.Kostovčík did formal analysis and writing—review and editing; M.Kolařík acquired the resources and did funding acquisition and writing—review and editing; D.H. did the investigation and writing—review and editing, P.D. did the conceptualization, supervision, project administration, and writing—review and editing.

### AUTHOR AFFILIATIONS

[1]Department of Biology and Ecology, Faculty of Science, University of Ostrava, Ostrava, Czechia

[2]Department of Zoology, Faculty of Science, Palacký University, Olomouc, Czechia

[3]Institute of Microbiology, Academy of Sciences of the Czech Republic, Prague, Czechia

## AUTHOR ORCIDs

Hana Šigutová http://orcid.org/0000-0003-1134-248X
Petr Pyszko http://orcid.org/0000-0002-3743-7201
Miroslav Kolařík https://orcid.org/0000-0003-4016-0335
Pavel Drozd http://orcid.org/0000-0002-4602-8856

## FUNDING

| Funder | Grant(s) | Author(s) |
| --- | --- | --- |
| Czech Science Foundation | GA22-29971S | Hana Šigutová |
| | | Petr Pyszko |
| | | Martin Šigut |
| | | Miroslav Kolařík |
| | | Denisa Hařovská |
| | | Pavel Drozd |

## AUTHOR CONTRIBUTIONS

Hana Šigutová, Funding acquisition, Writing – original draft | Petr Pyszko, Conceptualization, Formal analysis, Funding acquisition, Investigation, Methodology, Visualization, Writing – review and editing | Martin Šigut, Data curation, Investigation, Methodology, Writing – review and editing | Kateřina Czajová, Formal analysis, Investigation, Writing – review and editing | Martin Kostovčík, Formal analysis, Writing – review and editing | Miroslav Kolařík, Funding acquisition, Resources, Writing – review and editing | Denisa Hařovská, Investigation, Writing – review and editing | Pavel Drozd, Conceptualization, Funding acquisition, Project administration, Resources, Supervision, Writing – review and editing

## DATA AVAILABILITY

Raw demultiplexed sequencing data with sample annotations are available on the NCBI Bioproject website under accession number PRJNA932237.

## ADDITIONAL FILES

The following material is available online.

### Supplemental Material

**Figures S1 to S11, Table S1, S4 to S8 (Spectrum02994-23-s0001.docx).** Supplemental figures and tables.
**Table S2 (Spectrum02994-23-s0002.xlsx).** Overview of contaminant fungal ASVs.
**Table S3 (Spectrum02994-23-s0003.xlsx).** Overview of fungal taxa and number of reads.

### Open Peer Review

**PEER REVIEW HISTORY (review-history.pdf).** An accounting of the reviewer comments and feedback.

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
