## [Reviewer comments · Microbiology Spectrum]

Microbiology Spectrum

Concentration-dependent effect of plant secondary metabolites on bacterial and fungal microbiomes in caterpillar guts

Hana Šigutová, Petr Pyszko, Martin Šigut, Kateřina Czajová, Martin Kostovčík, Miroslav Kolařík, Denisa Višňovská, and Pavel Drozd

Corresponding Author(s): Petr Pyszko, Ostravska univerzita

Review Timeline:

Submission Date:	August 1, 2023
Editorial Decision:	August 30, 2023
Revision Received:	September 19, 2023
Editorial Decision:	September 29, 2023
Revision Received:	October 4, 2023
Accepted:	October 16, 2023

Editor: Rosario Gil

Reviewer(s): The reviewers have opted to remain anonymous.

Transaction Report:

DOI: <https://doi.org/10.1128/spectrum.02994-23>

August 30, 2023

Dr. Petr Pyszko
Ostravska univerzita
Ostrava
Czech Republic

Re: Spectrum02994-23 (Concentration-dependent effect of plant secondary metabolites on bacterial and fungal microbiomes in caterpillar guts)

Dear Dr. Petr Pyszko:

Thank you for submitting your manuscript to Microbiology Spectrum. It has been evaluated by two experts in the field, which posed some concerns that need to be addressed. When submitting the revised version of your paper, please provide (1) point-by-point responses to the issues raised by the reviewers as file type "Response to Reviewers," not in your cover letter, and (2) a PDF file that indicates the changes from the original submission (by highlighting or underlining the changes) as file type "Marked Up Manuscript - For Review Only". Please use this link to submit your revised manuscript - we strongly recommend that you submit your paper within the next 60 days or reach out to me. Detailed instructions on submitting your revised paper are below.

Link Not Available

Sincerely,

Rosario Gil

Journals Department
Reviewer comments:

Reviewer #1 (Comments for the Author):

Nice and well written study on the effect of secondary plant metabolites on caterpillar weight gain and gut microbiome composition.

suggestions for edits:

I would like to see in the sup materials the species specific data on weight gain and diversity for the treatments.

Consider adding that you only have two monophagous species? this would make it difficult to detect an effect of diet breadth

212: what is RDA?

249-255 and throughout: Make sure you refer to all figures and preferably plot a before b . in fig.1 is A weight gain on pure AD without PSM added? I would prefer a plot where you can see the individual datapoints in addition to mean {plus minus} SE

250: concentration of PSM, where larvae ...

In fig2 consider adding diversity in the diet alone also, since it's very high esp for fungi??

282-288: add a, b, c, to figure references and refer to all parts of the figure in the text

296: consider rephrasing eg: A small but significant amount of the variation in microbiome composition was explained by xxx (it is 1-8.5%)

322: unclear : In bacteria, feeding on AD without PSM and starving increased stochasticity of the community assembly while the low PSM concentration level decreased stochasticity, which then tended to increase with higher concentrations

361 an unexpected finding

385 and other places: specify that you mean host species

399 does not have to be an interaction, could simply be driven by the same thing such as diet diversity

423 was found in?

451-460: can you illustrate this in a figure?

Can you present Table S4 in a figure?

Fig. S1: you write "The rarefactions reaching mostly asymptotes show that the sequential depth was sufficient " but you should specify that you rarefy to 1000 and 200 and whether this is sufficient (looks like it is for bacteria, not clear for fungi)

Legend for Fig. S5 and S6 "with a significant trend with increasing PSMs concentration level" unclear. Also specify that it for artificial diet only

Reviewer #2 (Comments for the Author):

This work investigated the effect of plant secondary metabolites (PSM) on caterpillar gut bacterial and fungal microbiomes. Particularly, the authors emphasize the importance of considering PSM concentration and composition in understanding caterpillar-gut microbiome interactions. Despite the validation of the functional roles of identified microbial taxa and their significance for caterpillar hosts is missing, I think this paper is still interesting to the people in this research field, and deserves publication. However, I have some major concerns detailed below:

What do you mean "diet breadth"? Define it clearly at the beginning.

In the Importance, you mentioned "with low diversity and considerable variability", but in some cases, the gut microbiome diversity is not low, consider to revise this statement.

And what do you mean "the limited role of the microbiome in the plasticity of the herbivore diet." (Discussion L475 as well)? Considering that bacteria have diverse metabolic pathway and functional genes involved in degrading the toxic compound (functional redundancy), and that even the same species (but different strains) vary in genomic composition (different function genes), this conclusion is not supported by data from the current study, since it is based only on 16S and ITS marker gene sequencing, but not shotgun sequencing (such as metagenomics). This might also be the reason for "Analysis of KEGG pathways did not reveal any consistent pattern in bacterial metabolism related to PSM concentration," (L461).

A recent review (doi: 10.1146/annurev-ento-020723-102548), related with this direction, is recommended to be included in this manuscript. This review will strengthen the significance of the current work.

Fig. 1, how did you identify the nine different caterpillars? Was there any molecular methods applied here? Why don't you show us the "Impact on fitness" of other two groups, namely, starved caterpillars and oak leaf-fed caterpillars?

In the Materials and Methods, please detail the number of repetitions of different caterpillars.

For Microbial quantification, did you quantify both bacteria and fungi? The detailed primer pairs and QPCR settings/methods should be provided.

In L119, you mentioned that "Individuals from each species were weighed and reared on McMorran diet", so is this diet same as the "control artificial diet" you used in L108? Clarify this. Did the diet have a certain composition of plant material? I think the initial PSM in the control artificial diet should be considered.

In L122, you mentioned that "The medium concentration corresponded to the natural occurrence of these substances in tree foliage", the concentrations of T A and TV were 10-fold decreasing or increasing, but the concentration of SA was 3-fold, why?

In Fig. 3 b and d, I am confused that the two groups "Starved cater" and "Leaf-fed cater" have three treatment: LOW, MED, and HIGH. You mentioned that the diet was enriched with the PSM in different concentration, not "Starved cater" and "Leaf-fed cater", please explain.

Since you mentioned that "We hypothesized that microbiome of all species, regardless of diet breadth would contain tannin-degrading members as they were sampled from oaks", the tannin-degrading bacteria could be isolated to confirm this assumption.

Staff Comments:

Preparing Revision Guidelines

Please return the manuscript within 60 days; if you cannot complete the modification within this time period, please contact me. If you do not wish to modify the manuscript and prefer to submit it to another journal, please notify me of your decision immediately so that the manuscript may be formally withdrawn from consideration by Microbiology Spectrum.

13 September 2023
Dr Rosario Gil
Editor, Microbiology Spectrum

Dear Dr Gil,

Thank you for considering our manuscript entitled “**Concentration-dependent effect of plant secondary metabolites on bacterial and fungal microbiomes in caterpillar guts**” (02994-23) for publication. I, along with my co-authors, would like to re-submit its revised version.

We received two positive reviews on our manuscript. We carefully checked the manuscript and made appropriate changes in accordance with the reviewers’ suggestions. We believe that the comments of both reviewers have significantly improved the quality of our manuscript. Our responses are attached herewith.

We look forward to hearing from you regarding our submission. We would be glad to respond to any further questions and comments that you may have.

Sincerely,

Petr Pyszko

Reviewer #1 (Comments for the Author):

Nice and well written study on the effect of secondary plant metabolites on caterpillar weight gain and gut microbiome composition.

suggestions for edits:

I would like to see in the sup materials the species specific data on weight gain and diversity for the treatments.

Done (Fig. S2, S4, S5).

Consider adding that you only have two monophagous species? this would make it difficult to detect the effect of diet breadth

We appreciate your attention to this matter. Originally, our experimental design aimed to comprise three monophagous and six polyphagous species. However, we were able to gather a sufficient sample of 7/2 species, as monophagous species are inherently less common. Consequently, objective of the study has shifted subtly, and we have not emphasized the comparison between monophagous and polyphagous species in our final interpretation. Nevertheless, we believe it is still essential to include this comparison to provide a more comprehensive perspective due to the unique ecological dynamics of the two monophagous species.

212: what is RDA?

Redundancy analysis (information added, line 444).

249-255 and throughout: Make sure you refer to all figures and preferably plot a before b . in fig.1 is A weight gain on pure AD without PSM added? I would prefer a plot where you can see the individual datapoints in addition to mean {plus minus} SE

We added reference in the text to all subplots, as requested, and ordered them in alphabetical order. Regarding Fig. 1, information about weight gain on PSM-free artificial diet is not included.

Although we agree it might be beneficial, this would be based on a few individuals, and the readers can now find such information in Fig. S2 (created based on your first comment). In addition, our aim was to show how individual species responded to PSM (with the expectation that their ability to adapt to different PSM types and concentration would be species-specific, possibly different between monophagous and polyphagous), rather than to show how individual species gain on PSM-free diet. Regarding the plot with individual datapoints, we decided not to create such a plot as we believe that it would create a chaotic jumble of points with no additional information.

250: concentration of PSM, where larvae ...

Fixed (line 115).

In fig2 consider adding diversity in the diet alone also, since it's very high esp for fungi??

Thank you for this comment. This information is presented in Fig. S6.

282-288: add a, b, c, to figure references and refer to all parts of the figure in the text

Done (now line 120 and onwards).

296: consider rephrasing eg: A small but significant amount of the variation in microbiome composition was explained by xxx (it is 1-8.5%)

Thank you for your suggestion, rephrased (lines 148–149, and also 159).

322: unclear: In bacteria, feeding on AD without PSM and starving increased stochasticity of the community assembly while the low PSM concentration level decreased stochasticity, which then tended to increase with higher concentrations

Thank you for this comment; clarified (lines 171–175).

361: an unexpected finding

Corrected (line 208).

385: and other places: specify that you mean host species

Specified (lines 28, 220, 223, 227, 230, 234).

399: does not have to be an interaction, could simply be driven by the same thing such as diet diversity

We agree; we changed this statement slightly (line 247).

423: was found in?

Done (line 271).

451-460: can you illustrate this in a figure?

This part (now lines 298 and onwards) is based on the data coherently presented in tables S5 and S6. We believe that adding multiple plots wouldn't add any additional information value. We only present an overall trend of community assembly – differences between bacteria and fungi (Fig. S8).

Can you present Table S4 in a figure?

We have added Fig. S1 showing the difference of bacterial and fungal loads in caterpillar guts across different treatment types and PSM concentrations.

Fig. S1: you write "The rarefactions reaching mostly asymptotes show that the sequential depth was sufficient " but you should specify that you rarefy to 1000 and 200 and whether this is sufficient (looks like it is for bacteria, not clear for fungi)

(Now Fig. S3). Here, rarefactions were used only to illustrate that the sequential depth was sufficient. However, in further analyses, we used mainly unrarefied values (e.g., for RDA or analyses of similarity). Rarefied values (to 1000 reads for bacteria and 200 for fungi) were used only when analyses would be distorted when using unrarefied values (null models and comparison of richness). Therefore, rarefied bacterial and fungal richness should be perceived rather as a diversity index which does not reflect a real richness and is not related to this figure.

Legend for Fig. S5 and S6 "with a significant trend with increasing PSMs concentration level" unclear. Also specify that it for artificial diet only

Clarified (now Fig. S9 and S10). Regarding the trends in artificial diet only, we tested for them using the same methodology (threshold indicator taxa analysis with the same cutoff values for purity and reliability), and there were no significant trends, neither for bacteria, nor for fungi.

Reviewer #2 (Comments for the Author):

This work investigated the effect of plant secondary metabolites (PSM) on caterpillar gut bacterial and fungal microbiomes. Particularly, the authors emphasize the importance of considering PSM concentration and composition in understanding caterpillar-gut microbiome interactions. Despite the validation of the functional roles of identified microbial taxa and their significance for caterpillar hosts is missing, I think this paper is still interesting to the people in this research field, and deserves publication. However, I have some major concerns detailed below:

What do you mean "diet breadth"? Define it clearly at the beginning.

We added a definition, not at the first mention in the abstract (due to the word number limitation), but in Introduction (lines 81–83). Nevertheless, in abstract, the expression "diet breadth" is followed by the explanation that the experiment comprised monophagous and polyphagous caterpillars. Therefore, we believe that it will be clear for the readers, even without reading the definition in the Introduction.

In the Importance, you mentioned "with low diversity and considerable variability", but in some cases, the gut microbiome diversity is not low, consider to revise this statement.

Thank you for this comment; we agree. We made a slight change to this sentence to reflect there may be some exceptions (line 40).

And what do you mean "the limited role of the microbiome in the plasticity of the herbivore diet." (Discussion L475 as well)? Considering that bacteria have diverse metabolic pathway and functional genes involved in degrading the toxic compound (functional redundancy), and that even the same species (but different strains) vary in genomic composition (different function genes), this conclusion is not supported by data from the current study, since it is based only on 16S and ITS marker gene sequencing, but not shotgun sequencing (such as metagenomics). This might also be the reason for "Analysis of KEGG pathways did not reveal any consistent pattern in bacterial metabolism related to PSM concentration," (L461).

Thank you for this comment. The statement "Our study revealed the lack of differences in the PSM-induced responses of microbial assemblages between monophagous and polyphagous species, suggesting the limited role of the microbiome in the plasticity of the herbivore diet" is related to the

sentence in introduction: "The PSM-induced changes in the gut microbiome structure may be considered as initial step towards host plant specialization. Investigating changes in the gut microbiome of polyphagous and monophagous species under varying concentrations of PSM native and non-native to host plants may help clarify the role of the microbiome in herbivore diet plasticity, and, ultimately, host plant specialization". We believe that from this point of view, our conclusion is supported by the results. By the "limited role of the microbiome in the plasticity of herbivore diet" we mean simply the fact that we didn't find any significant differences in reaction of the microbiomes to plant secondary metabolites between monophagous and polyphagous caterpillar species, regardless of whether those metabolites were native to their host plants or not.

A recent review (doi: 10.1146/annurev-ento-020723-102548), related with this direction, is recommended to be included in this manuscript. This review will strengthen the significance of the current work.

Thank you for your suggestion, we included the recommended review in our manuscript (ref. 15).

Fig. 1, how did you identify the nine different caterpillars? Was there any molecular methods applied here?

All caterpillar species used in this study are morphologically distinct, except of Agriopsis aurantiaria which can be confused with another Agriopsis species. However, our team has been involved in caterpillar studies for over 10 years now, including an extensive field sampling (e.g., <https://doi.org/10.1111/1365-2656.12646>; <https://doi.org/10.1002/ece3.4194>; <https://doi.org/10.1093/femsec/fiaa116>; <https://doi.org/10.1002/ece3.7005>; <https://doi.org/10.1038/s41598-022-19855-5>; <https://doi.org/10.1128/spectrum.03160-22>). Therefore, our team members have a strong expertise in morphological identification of the focal caterpillar species, and there was no need to apply any molecular methods.

Why don't you show us the "Impact on fitness" of other two groups, namely, starved caterpillars and oak leaf-fed caterpillars?

Thank you for this comment; very relevant. Starved caterpillars had always decreasing trend in weight; therefore, we do not consider as important to show them in the figure (but based on your comment, we added this information to results, line 112). Leaf-fed caterpillars were all weighted, and part of them (used as a leaf-fed control group) was killed instantly, without further rearing. Therefore, they were not reared on oak leaves, simultaneously with AD-fed caterpillars. We added this information to the MS, and also clarified the role of the other control groups (lines 355–359). Therefore, weight gain of leaf-fed individuals couldn't have been detected, based on our experimental design. We are aware of the possible flaws of this experimental design. Nevertheless, variability in the leaf microbiota is (also based on our previous studies) so high that we considered the comparison of caterpillars reared on AD with caterpillars further fed with fresh oak leaves to be less relevant than the use of caterpillars fed on the same leaf food (i.e., sampled in the field), in which only a part of them were further fed with AD. Therefore, it can be assumed that at the start of the experiment the caterpillars sampled in the field from oaks (i.e., fed by oak leaves) and killed were the best possible comparison group to the caterpillars further fed with AD, and that during the experiment, no significant trend would have occurred for them (assuming that they wouldn't have been fed with new leaves all the time).

In the Materials and Methods, please detail the number of repetitions of different caterpillars.

List of caterpillar species with number of repetitions across various types of PSMs and concentrations is presented in Table S1. Because the number of repetitions across types and concentrations of PSMs was not always equal, we believe that this information should be presented rather in the Supplementary material than in the manuscript.

For Microbial quantification, did you quantify both bacteria and fungi? The detailed primer pairs and QPCR settings\methods should be provided.

As stated in the M&M section (line 385 and onwards), we quantified both bacteria and fungi using Femto bacterial and fungal kit, respectively. According to manufacturer's manuals (https://files.zymoresearch.com/protocols/e2006_femto_bacterial_dna_quantification_kit_ver.pdf; https://files.zymoresearch.com/protocols/e2007_femto_fungal_dna_quantification_kit.pdf), there are no information about primer pairs used in the kit; there are only the thermocycling parameters. We feel that providing only this incomplete information is redundant and don't fit in the manuscript.

In L119, you mentioned that "Individuals from each species were weighed and reared on McMorran diet", so is this diet same as the "control artificial diet" you used in L108? Clarify this. Did the diet have a certain composition of plant material? I think the initial PSM in the control artificial diet should be considered.

Clarified (lines 346-347; and 355-359). McMorran diet was prepared exactly according to the paper cited, and from the plant material, it contains only wheat germs which should be free from PSM.

In L122, you mentioned that "The medium concentration corresponded to the natural occurrence of these substances in tree foliage", the concentrations of T A and TV were 10-fold decreasing or increasing, but the concentration of SA was 3-fold, why?

Thank you for this comment. The concentration gradient was established as 10-fold in tannin and tannivin but only 3-fold in salicylate due to practical reasons; after adding SA in the concentration above 1%, the diet didn't solidify and remained liquid. The concentration gradient was therefore adjusted in a way that the diet had always similar consistency, regardless of the substance added, to ensure that it won't be rejected by caterpillars due to the unsuitable consistency.

In Fig. 3 b and d, I am confused that the two groups "Starved cater" and "Leaf-fed cater" have three treatment: LOW, MED, and HIGH. You mentioned that the diet was enriched with the PSM in different concentration, not "Starved cater" and "Leaf-fed cater", please explain.

Thank you for this comment; we understand the confusion. This figure shows a comparison of the gut bacterial and fungal composition of caterpillars fed with LOW, MEDIUM and HIGH-PSM concentration AD with composition of starved and leaf-fed caterpillars. CON column shows a comparison with control AD (without PSM). We adjusted the figure caption to avoid misinterpretation (lines 805-808).

Since you mentioned that "We hypothesized that microbiome of all species, regardless of diet breadth would contain tannin-degrading members as they were sampled from oaks", the tannin-degrading bacteria could be isolated to confirm this assumption.

Thank you for this comment. In our hypothesis, we didn't assume the occurrence of the specific symbionts, but rather that the whole microbiome would be adjusted to digest tannins. Therefore, our manuscript aimed to examine these relationships (from simple richness to networks) and how they change with other substances - for example, they do not change in the presence of tannin, but they do in the presence of salicylate (although the "members" remain similar). We adjusted that hypothesis slightly to fit in our aims (lines 99-100).

September 29, 2023

Dr. Petr Pyszko
Ostravska univerzita
Ostrava
Czech Republic

Re: Spectrum02994-23R1 (Concentration-dependent effect of plant secondary metabolites on bacterial and fungal microbiomes in caterpillar guts)

Dear Dr. Petr Pyszko:

Thank you for submitting your revised manuscript to Microbiology Spectrum. As you will see your paper is very close to acceptance.

I have confirmed myself that you have followed the reviewers' recommendations and argued adequately to their concerns. However, I want to make a couple of minor suggestions that I would like you to address before the manuscript is ready for publication.

First one refers to the bar graphs presented in the figures, mostly as supplementary material. It would be convenient to add asterisks indicating the statistical significance of the observed differences, if any. Second one is regarding your answer to the Reviewer #2 comment on the reason to select a specific concentration gradient for the different substances of interest found in the tree foliage (L122 in the previous version of the manuscript). I think this practical reason to select a specific concentration gradient should be added to the text, as it could help other authors that intend to do similar studies.

When submitting the revised version of your paper, please provide (1) responses to my comments directly in your cover letter, if needed, and (2) a PDF file that indicates the changes from the original submission (by highlighting or underlining the changes) as file type "Marked Up Manuscript - For Review Only". Please use this link to submit your revised manuscript - we strongly recommend that you submit your paper within the next 60 days or reach out to me. Detailed instructions on submitting your revised paper are below.

Link Not Available

Sincerely,

Rosario Gil

Journals Department
Reviewer comments:

Staff Comments:

Preparing Revision Guidelines

Please return the manuscript within 60 days; if you cannot complete the modification within this time period, please contact me. If you do not wish to modify the manuscript and prefer to submit it to another journal, please notify me of your decision immediately so that the manuscript may be formally withdrawn from consideration by Microbiology Spectrum.

We received only editor's comments. The responses are below.

Editor's comments:

I have confirmed myself that you have followed the reviewers' recommendations and argued adequately to their concerns. However, I want to make a couple of minor suggestions that I would like you to address before the manuscript is ready for publication.

First one refers to the bar graphs presented in the figures, mostly as supplementary material. It would be convenient to add asterisks indicating the statistical significance of the observed differences, if any.

- *Thank you for this comment. We considered your request regarding adding statistical significance of the observed differences to the barplots in the manuscript and also in the Supplementary material. To provide additional information about the significance of differences between the individual columns in the bar plots, we utilized a system of labeling significance with letters – individual letters indicate groups that are significantly different from each other. These significances are provided for each level of the main factor displayed on the x-axis. We believe that this approach emphasizes the narrative we want to convey (after careful consideration, we decided not to compare every group with every other group, groups divided by both factors, as such representation seemed to us to further complicate and hinder orientation in the results). We include this information in each of the plots, and we added the explanation to the figure legends (line 800 and onwards). The only exception is Figure 3 (a and c), where it made perfect sense to display the differences between individual levels of the factor shown on the x-axis for groups divided according to the legend. However, even in this case, we did not attempt to compare all groups with each other, as it lacked meaning.*

Second one is regarding your answer to the Reviewer #2 comment on the reason to select a specific concentration gradient for the different substances of interest found in the tree foliage (L122 in the previous version of the manuscript). I think this practical reason to select a specific concentration gradient should be added to the text, as it could help other authors that intend to do similar studies.

- *We added an explanation of the secondary metabolite concentration gradient used in the feeding experiment (line 347–351).*

October 5, 2023

Dr. Petr Pyszko
Ostravska univerzita
Ostrava
Czech Republic

Re: Spectrum02994-23R2 (Concentration-dependent effect of plant secondary metabolites on bacterial and fungal microbiomes in caterpillar guts)

Dear Dr. Petr Pyszko:

Your manuscript has been accepted, and I am forwarding it to the ASM Journals Department for publication. You will be notified when your proofs are ready to be viewed.

Sincerely,

Rosario Gil
Editor, Microbiology Spectrum
